# SP and KLF Transcription Factors in Cancer Metabolism

**DOI:** 10.3390/ijms23179956

**Published:** 2022-09-01

**Authors:** Emilia J. Orzechowska-Licari, Joseph F. LaComb, Aisharja Mojumdar, Agnieszka B. Bialkowska

**Affiliations:** Department of Medicine, Renaissance School of Medicine at Stony Brook University, Stony Brook, New York, NY 11794, USA

**Keywords:** specificity proteins, Krüppel-like transcription factors, cancer, metabolism

## Abstract

Tumor development and progression depend on reprogramming of signaling pathways that regulate cell metabolism. Alterations to various metabolic pathways such as glycolysis, oxidative phosphorylation, lipid metabolism, and hexosamine biosynthesis pathway are crucial to sustain increased redox, bioenergetic, and biosynthesis demands of a tumor cell. Transcription factors (oncogenes and tumor suppressors) play crucial roles in modulating these alterations, and their functions are tethered to major metabolic pathways under homeostatic conditions and disease initiation and advancement. Specificity proteins (SPs) and Krüppel-like factors (KLFs) are closely related transcription factors characterized by three highly conserved zinc fingers domains that interact with DNA. Studies have demonstrated that SP and KLF transcription factors are expressed in various tissues and regulate diverse processes such as proliferation, differentiation, apoptosis, inflammation, and tumorigenesis. This review highlights the role of SP and KLF transcription factors in the metabolism of various cancers and their impact on tumorigenesis. A better understanding of the role and underlying mechanisms governing the metabolic changes during tumorigenesis could provide new therapeutic opportunities for cancer treatment.

## 1. Specificity Proteins (SPs) and Krüppel-like Factors (KLFs)

Tumorigenesis originates as a consequence of oncogenic and tumor suppressor mutations [1]. The outcome of these modifications impacts metabolic pathways. It allows tumor cells to thrive in a nutrient-limited environment and affects proliferation, survival, resistance, motility, and invasiveness, to mention a few [2,3,4,5]. Specificity proteins (SP) and Krüppel-like factors (KLF) belong to a highly conserved family of transcription factors characterized by three zinc finger domains localized to the C-terminus of the proteins (Figure 1) [6,7,8,9]. Surprisingly, these domains are the sole portions showing a significant level of similarity throughout the entire SP/KLF family. Furthermore, motifs within the N-terminal and middle portions of these proteins regulate the interaction between proteins and the selectivity of their interaction with DNA and show a high level of diversity [6,10,11,12,13]. Nine SPs and eighteen KLFs have been identified in multiple species [7,9,14,15,16,17,18,19]. To demonstrate relationship between these factors, we included a phylogenetic tree based on the consensus of the sequences of the zinc-finger domains in SP/KLF proteins identified in primates [human (*Homo sapiens*), chimpanzee (*Pan troglodytes*), orangutan (*Pongo abelii*), macaque (*Macaca mulatta*), lemur (*Microcebus murinus*)] and rodents [naked mole rat [nmr] (*Heterocephalus glaber),* rat (*Rattus norvegicus*), and mouse (*Mus musculus*)] demonstrating the evolutionary relationship of these proteins (Figure 2) [20]. SP/KLF proteins’ role in the regulation of signaling pathways in embryogenesis, development, disease progression, and carcinogenesis on molecular and cellular levels has been extensively studied and reviewed [6,7,8,16,21,22,23,24,25]. Their functions as tumor suppressors or oncogenes are well-defined in multiple types of cancers [26]. Importantly, their role in the transcriptional regulation of pathways involved in the metabolism of glucose, lipids, and amino acids under homeostatic conditions has been extensively discussed [27,28]. KLF4 has been shown to promote nutrient uptake while KLF7 decreases it. KLF6, KLF11, and KLF15 participate in glucose uptake and/or regulation of glucose metabolism [12,27]. KLF10, KLF14, and KLF15 regulate gluconeogenesis by activating phosphoenolpyruvate carboxykinase (PEPCK), while KLF11 suppresses PEPCK activity [29,30,31,32]. KLF15 activates mTORC1 signaling to promote fatty acid oxidation [12]. Multiple KLFs (3, 4, 5, 7, 9, and 15) play important role in regulation of adipogenesis [12,28,33]. It has been shown that SP1 modulation of leptin during homeostasis is activated by insulin-stimulated glucose metabolism [34] and SP1 regulates fatty acid metabolism [35]. Furthermore, there are excellent comprehensive reviews focused on KLFs’ role in regulating metabolic processes in fatty liver disease, obesity, or cardiac diseases [12,28]. Acknowledging the profound role of SP/KLF factors in metabolism regulation under homeostatic conditions, their involvement in tumorigenesis, and the latest efforts to target their function, we present a review concerning their involvement in the regulation of cancer metabolism.

## 2. Gastrointestinal Tract

The human gastrointestinal (GI) tract is a part of the digestive system whose primary functions are digestion, absorption of nutrients, and excretion of the waste products of digestion [37,38]. The GI tract is often described as a long tube composed of multiple organs and is conventionally divided into the upper and lower GI tracts. The upper GI tract starts with the mouth and transits into the esophagus, stomach, duodenum, jejunum, and ileum, while the lower GI tract consists of the colon, rectum, and anus. Proper functioning of the GI tract assures the body’s homeostasis. However, disorders including alteration of the cellular metabolism occur, especially during carcinogenesis [39]. Gastrointestinal cancers include esophageal, gastric, colon, and rectal tumors [40]. As reported by the WHO, in 2020, colorectal and stomach cancers are the third (10.0% cases) and fifth (5.6%) most commonly diagnosed cancers, respectively. At the same time, colorectal cancer is the second leading cause of cancer death (9.4%), followed by liver (8.3%) and stomach cancers (7.7%) [41].

**Figure 2 ijms-23-09956-f002:**
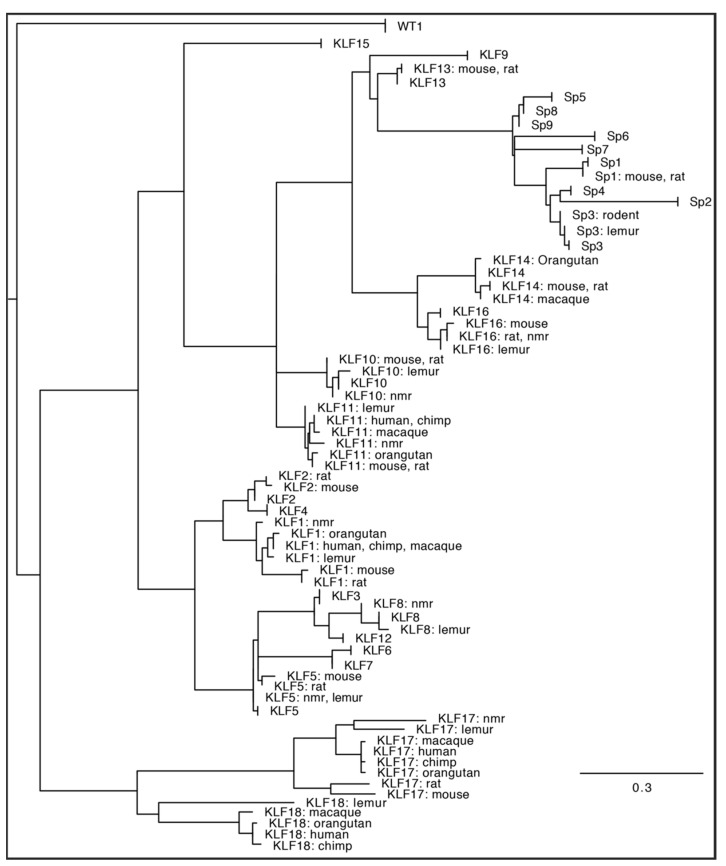
Maximum likelihood phylogeny of the DNA binding domain [15] from all KLF-SP family members identified from human (*Homo sapiens*), chimpanzee (*Pan troglodytes*), orangutan (*Pongo abelii*), macaque (*Macaca mulatta*), lemur (*Microcebus murinus*), naked mole rat [nmr] (*Heterocephalus glaber)*, rat (*Rattus norvegicus*), and mouse (*Mus musculus*) genome sequence databases. Respectively, those genome database builds were: GRCh38.p14, Clint_PTRv2, Susie_PABv2, Mmul_10, Mmur_3.0, HetGla_female_1.0, mRatBN7.2, and GRCm39. Lemur KLF2 and KLF15 are absent from the Mmur_3.0 database, while the sequence RPGA01000022.1 was used to repair a sequencing gap in the naked mole rat reference genome SP4 sequence, and RGSC6.0/rn6 was used to repair the rat KLF13 sequence. The DNA binding domain of Wilms’ Tumor 1 (WT1) is included as an outgroup. Consensus phylogeny was produced in Geneious 10.2.6 using RAxML 8.2.11 [42] after sampling 200 bootstrap replicate trees calculated using the “Gamma GTR” protein evolutionary model. The tree was rooted on the WT1 outgroup. For display, branches with no distance (100% amino acids identity) were collapsed. When >50% of the taxa for a given gene were collapsed, all species information is excluded from the annotation. Otherwise, the annotation includes all species represented by a given branch. The scale bar represents amino acid substitutions/site.

### 2.1. Esophageal Cancer

Esophageal cancer is classified as either squamous cell carcinoma (ESCC) or, more frequently, diagnosed as adenocarcinoma [43]. In addition, lipid metabolism is often altered in esophageal cancer [44].

#### Lipid Metabolism Alteration

In ESCC, a high level of lysophosphatidylcholine acyltransferase 1 (LPCAT1) expression promotes the activation of phosphatidylinositol 3-kinases (PI3K). This, in turn, results in the incorporation of specific protein 1 (SP1) and sterol regulatory-element binding protein 1 (SREBP-1) into the nucleus, influencing the expression of squalene monooxygenase (SQLE), one of the key enzymes of cholesterol biosynthesis (Figure 3(1A)). This increased cholesterol synthesis de novo is crucial for cancer development [45]. In esophageal adenocarcinoma, KLF5 was identified as one of the four esophageal adenocarcinoma-specific master regulator transcription factors (MRTF) that control, among others, de novo synthesis of fatty acids, phospholipids, and sphingolipids through peroxisome proliferator-activated receptor gamma (PPARγ) [43]. It was shown that three MRTFs (including KLF5) co-occupied the enhancer region, and KLF5 alone binds to the *PPARG* promoter. PPARγ upregulates genes involved in de novo fatty acid, phospholipids, and sphingolipids synthesis. Additionally, a transcriptional feedback loop between MRTFs and fatty acid synthesis has been characterized, revealing that MRTFs activate each other through the nuclear receptor, PPARγ (Figure 3(1B)). Importantly, in vivo experiments strongly suggest that a high-fat diet (HFD) promotes esophageal adenocarcinoma growth through activating PPARγ. In xenograft samples, the expression levels of both MRTFs and canonical PPARγ target genes were upregulated by HFD [43].

### 2.2. Gastric Cancer

In gastric cancer, glucose metabolism and Ca^2+^ intracellular concentration alterations were the main metabolism-related changes [46,47,48].

#### 2.2.1. Glucose Metabolism Alteration

KLF8 was recently shown to be upregulated in many gastric cancer patient samples, which correlated with a higher maximum standardized uptake value (SUVmax), increased glucose utilization, lactate concentrations, and ATP production indicating the Warburg effect. Potentially, aerobic glycolysis in gastric cancer is promoted via targeting the *GLUT4* promoter by KLF8 [49] (Figure 3(2A)). Similarly, KLF12 was shown to be upregulated and be a positive regulator of glucose uptake, lactate and ATP production, and hexokinase 2 (HK-2) protein levels in AGS and SNU-638 gastric cancer cell lines. Mechanistically, the proto-oncogenic effect of KLF12 is achieved by the downregulation of miR-876-3p, the direct negative regulator of KLF12, which is extensively absorbed by the increased level of the circular RNA circ-RNF111 [50] (Figure 3(2B)).

#### 2.2.2. Intracellular Calcium Concentration Alteration

KLF4 is believed to act as a tumor suppressor, and its low expression is negatively associated with the overall survival rate in gastric cancer patients [51]. One of the mechanisms by which KLF4 operates is through the regulation of intracellular Ca^2+^ concentration (Figure 3(2C)). The loss of Ca^2+^ homeostasis affects many cellular processes associated with tumor development [52]. It was shown that together with SP1 and SP3, KLF4 binds directly to ATPase sarcoplasmic/endoplasmic reticulum Ca^2+^ transporting 3 (ATP2A3) proximal promoter elements and regulates its expression during epithelial cancer cell differentiation. ATAP2A3 encodes the sarco/endoplasmic reticulum Ca^2+^ -ATPase 3 (SERCA3), an isoform that is expressed among others in gastric and colonic epithelial cells. The high-grade tumors correlated with an undifferentiated phenotype and less abundance of SERCA3 expression [53].

### 2.3. Colorectal Cancer

In colorectal cancer, members of the Krüppel-like family transcription factors act either as activators or suppressors of tumorigenesis by altering the primary energy production associated pathways: glucose and lipids metabolism [54]. Moreover, extrinsic factors affecting metabolic phenotype have been described.

#### 2.3.1. Glucose Metabolism Alteration

KLF4 serves as a tumor suppressor by turning glucose metabolism into less Warburg-like, but its levels are frequently decreased in colorectal tumors [51,55]. KLF4 increases the expression of essential glycolytic proteins, among others, hexokinase 2 (HK-2) and pyruvate kinase M2 (PKM2), and lactate export protein monocarboxylate transporter 4 (MCT4), which favors oxidative phosphorylation over lactate accumulation. Additionally, even though KLF4 does not increase the expression levels of glucose transporter (GLUT1) mRNA and protein, it increases the translocation of GLUT1 to the cellular membrane (Figure 3(3A)). Lastly, KLF4 induces autophagy in case of severe glucose starvation and reduces stress-induced ROS within cells [56].

Some metabolic changes are induced by environmental influences, for instance, microbiota. Recent studies showed that the SP1 transcription factor increases glycolysis, lactate production, and glucose uptake by targeting the *Fusobacterium nucleatum*-induced axis of long non-coding RNA (lncRNA) enolase1-intronic transcript 1 (*ENO1-IT1*) and KAT7 histone modification. *F. nucleatum* upregulates lncRNA ENO1-IT1 transcription by increasing the binding efficiency of transcription factor SP1 to the promoter region of lncRNA ENO1-IT1. Then, elevated lncRNA ENO1-IT may recruit KAT7 histone acetyltransferase to the promoter of the ENO1 gene and regulate ENO1 transcription via epigenetic modulation. Notably, an elevated abundance of *F. nucleatum* corresponds with elevated ENO1-IT1, ENO1, and poor patient outcome prognosis (Figure 3(3B)) [57].

KLF14 was shown to regulate glycolysis by downregulating lactate dehydrogenase B (*LDHB*) on a transcriptional level. However, in colorectal cancer cells, *KLF14* expression is downregulated, and as a result, the overall glycolysis rate increases via intensified glucose uptake, lactate and ATP production [58] (Figure 3(3C)). The possible mechanism underlying KLF14 downregulation in colorectal cancer cells is based on the interaction occurring between circTADA2A and its target miR-374a-3p. Usually, circTADA2A keeps the level of KLF14 “high”, but since it is downregulated, miR-374a-3p is upregulated and downregulates KLF14 [59] (Figure 3(3C)).

#### 2.3.2. Lipid Metabolism Alteration

Alteration of lipid metabolism in colorectal cancer influences de novo cholesterol synthesis or increases lipids catabolism [60,61,62].

KLF13, for instance, was shown to act as a tumor suppressor by binding to the 3-hydroxy-3-methyl-glutaryl coenzyme A synthase 1 (*HMGCS1*) promoter, inhibiting its expression and thereby inhibiting cholesterol biosynthesis (Figure 3(3D)) [63]. KLFs may also reduce cholesterol synthesis by mediating the antitumor potential of drugs. Recently, KLF2 was shown to mediate the antitumor potential of simvastatin, a 3-hydroxy-3-methyl-glutaryl coenzyme A (HMG-CoA) reductase inhibitor. A mutated variant of p53 observed in about 50% of colon cancer cases upregulates cholesterol synthesis and downregulates KLF2 and p21^WAF1/CIP1^ levels (Figure 3(3E)). Simvastatin significantly upregulated KLF2 and p21^WAF1/CIP1^ expression levels in SW1116 colon cancer cells carrying a mutated variant of p53 but not in p53 wild-type HCT116 cells. Thus, KLF2 upregulates p21^WAF1/CIP1^ and downregulates the expression levels of a mutated variant of p53, reducing cholesterol synthesis [64].

KLF9 and SP5 were recently linked as potential mediators of malic enzyme 1 (ME1)-induced tumorigenesis. ME1 is a cytosolic enzyme that links the catabolic glycolysis pathways and the Krebs cycle to the anabolic pathways of fatty acid and cholesterol biosynthesis through NADPH (Figure 3(3F)). *Apc^Min/+^*/*ME1-Tg* mice showed increased *Sp5* transcript level and KLF9 nuclear staining within the crypts and villi lamina propria, which corresponded with significant and greater numbers of adenomas in the small intestine (jejunum and ileum) as compared to *Apc^Min/+^* mice [65].

Another example of a microenvironmentally derived factor affecting metabolism is the effect of a high-fat diet (HFD) on hormone-sensitive lipase (HSL). HFD stimulates SP1-dependent β2-adrenergic receptor (β2AR) expression, leading to increased HSL phosphorylation at S552 via the cAMP/PKA axis (Figure 3(3G)). Activating HSL by β-adrenergic stimulation reduces triglycerides levels, increases free fatty acids levels, increases β-oxidation gene expressions, and increases ATP production [66].

## 3. Liver Cancer

The Warburg shift and upregulation of lipid catabolism are known characteristics of metabolic changes in liver cancers [67,68].

### 3.1. Aberrant Lipid Metabolism

Increased de novo lipogenesis promotes cell proliferation [69]. Furthermore, dysregulation of enzymes involved in lipid metabolism indicates poor prognosis in patients with hepatocellular carcinoma (HCC) [70,71]. The KLF2/Peroxisome proliferation-activated receptor γ (PPARγ) axis is perturbed by aberrant upregulation of ornithine decarboxylase 1 (ODC1) in oncogenic hepatic cells. In vitro *ODC1* silencing has been shown to restore KLF2 expression, downregulating PPARγ and reestablishing homeostatic rates of lipogenesis and glucose transport. Additionally, *ODC1* silencing suppressed tumorigenesis of HCC xenografts in vivo [72] (Figure 4A). KLF4 has been characterized as having tumor-suppressing activities in many cancers, including gastric, colorectal, liver, and pancreatic cancer as illustrated herein. It has been shown that KLF4 expression is suppressed in HCC [73,74]. This results in decreased transcription of monoglyceride lipase (*MGLL*) and dysregulation of lipolysis. In vitro *KLF4* overexpression in HCC cells restores homeostatic lipid metabolism and decreases tumorigenesis and cell migration by positively regulating MGLL expression [75] (Figure 4B). KLF13 has various roles (and dysregulated expression levels) across many cancers. As discussed in this review, KLF13 suppresses cellular proliferation in colorectal cancer via transcriptional inhibition of HMGCS1-mediated cholesterol synthesis [63]. In HCC, however, overexpressed *KLF13* transcriptionally promotes Acyl-CoA thioesterase 7 (*ACOT7*) [76]. Elevated ACOT7 upregulates long-chain acyl-CoA metabolism to free monounsaturated fatty acid and CoA, resulting in an accumulation of C18:1 oleic acid. Furthermore, oleic acid production increases cell proliferation and migration. Likewise, *KLF13* silencing reduced *ACOT7* expression, and in vitro downregulation of *ACOT7* in *KLF13*-overexpressing cells reduced proliferation and cell migration (Figure 4C). Collectively, KLF13 and ACOT7 were both demonstrated to have oncogenic roles in HCC [76].

### 3.2. Aberrant Glycolytic Metabolism

Oncogenic cells preferentially utilize aerobic glycolysis for ATP production in the Warburg effect [77]. SET8, the monomethyltransferase of histone 4 at lysine 20, is implicated in many biological processes, including glucose metabolism. It binds to and inactivates KLF4, which in turn downregulates *SIRT4*, a mitochondrial sirtuin involved in glutamine and fatty acid metabolism [78] (Figure 4D). Decreased SIRT4 expression shunts cellular metabolism away from oxidative phosphorylation by inhibiting both the conversion of glutamate to α-ketoglutarate by glutamate dehydrogenase (GLUD1) and pyruvate to acetyl-CoA by pyruvate dehydrogenase (PDH). Thus, HCC cells preferentially use lactate-producing aerobic glycolysis for cellular metabolism. Furthermore, in vitro restoration of KLF4 returned oxidative phosphorylation to normal levels from aerobic glycolysis. Proliferation and migration were reduced, and apoptosis increased in KLF4-expressing HCC cells [78]. Taken together, SET8 may serve as a master regulator of the KLF4/SIRT4 axis responsible for the shift from oxidative phosphorylation to aerobic glycolysis in HCC.

The first step of glycolysis is the phosphorylation of glucose to glucose-6-phosphate by the family of hexokinases. Specifically, hexokinase type II (HK2) is upregulated more than 100-fold in proliferating cancer cells compared to normal cells. SP1, SP2, and SP3 bind to 2 or more of 4 GC boxes within the *HK2* promoter region of rat hepatoma cells under tumor-favorable conditions (i.e., serum and glucose-supplemented media) [79]. Furthermore, glucose increases dephosphorylated SP1 levels which have greater DNA binding affinity and promoter activity of the glycolytic enzymes aldolase A and pyruvate kinase [80]. Aldolase A catalyzes fructose-1,6-bisphosphate to glyceraldehyde 3-phosphate and dihydroxyacetone phosphate, whereas pyruvate kinase catalyzes phosphoenolpyruvate to pyruvate in the fourth and eighth steps of glycolysis, respectively. Conversely, in vitro glucose deprivation reduces SP1-depedent transcription of these two enzymes [80]. This results in decreased cellular glycolysis and proliferation.

## 4. Pancreas

Pancreatic cancer also conforms to the Warburg effect [81]. Lactate dehydrogenase (LDH) catalyzes pyruvate to lactate in the final step of aerobic glycolysis [82]. LDH has five tetrameric isozymes (LDH1-5) composed of M and/or H subunits. LDH5, comprised of 4 M subunits, is the most effective isozyme in catalyzing pyruvate to lactate, and increased expression is an attribute of many metastatic cancers. Lactate dehydrogenase A (LDHA), the gene responsible for transcribing M subunits, is upregulated in pancreatic cancer [83].

KLF7 promotes the expression of several glycolysis-related proteins, including HK2, PFKBF3, and PDK1 [84]. A miR-185-5p/KLF7 axis exists in pancreatic cancer such that downregulated miR-185 results in decreased tumor suppression, and increased KLF7 drives oncogenic cytokine production, proliferation, and migration. Additionally, the expression of the long intergenic non-protein coding RNA LINC00152 is increased in pancreatic cancer. LINC00152 silencing significantly lowered glucose consumption, lactic acid production, cellular ATP levels, and glycolysis-related enzyme expression. Furthermore, LINC00152 binds to and negatively regulates miR-185-5p. In vitro and in vivo deletion of LINC00152 upregulates miR-185-5p, and overexpression of miR-185-5p, in turn, suppresses glycolysis. miR-185-5p binding at the 3’ UTR of *KLF7* causes downregulation of *KLF7*. LINC00152 silencing is associated with the downregulation of *KLF7* via miR-185-5p and subsequent downstream suppression of glycolysis (Figure 5A) [84].

In addition to inhibiting proliferation, differentiation, and migration, KLF4 has been implicated in altered cellular metabolism. Recently, KLF4 was identified as a transcriptional regulator of *LDHA* (Figure 5B). Furthermore, in vitro and in vivo overexpression of *KLF4* negatively regulated *LDHA*, and *KLF4* silencing resulted in increased *LDHA* levels [85]. KLF10 plays a role in regulating glucose metabolism. KLF10 downregulation in many cancers promotes increased epithelial-mesenchymal transition (EMT) and glycolysis [86]. Deletion of *KLF10* accelerates pancreatic adenocarcinoma (PDAC) progression [87]. Sirtuin 6 (SIRT6) also plays a role in regulating glucose metabolism [88]. KLF10 binds to and activates SIRT6 [86]. Downregulating *KLF10* decreased SIRT6 expression and led to an increased glycolytic activity [86] (Figure 5C). In vitro overexpression of *KLF10* reduced glycolytic enzyme levels, lactate production, and mitochondrial oxidative phosphorylation via SIRT6. In contrast, glycolysis and EMT were upregulated by *KLF10* knockdown-mediated aberrant expression of NFκB and HIF1α. Overexpressing *SIRT6* in *KLF10* knockout cells decreased glycolytic activity. Linoleic acid (LA) increased *SIRT6* expression and restored normal glucose metabolism in vitro, and increased survival in vivo. Both genetic modulation and pharmacological intervention indicate that KLF10 positively regulates SIRT6 [86].

## 5. Breast Cancer

Breast cancer is a leading cause of death in females [89,90]. During breast cancer development and progression, there is an increase in the metabolism of glucose, lipids, and amino acids [91]. In breast cancer, there is often a shift toward glycolytic metabolism to supply the cell with higher amounts of ATP [92]. This increase in glycolysis is accompanied by the accumulation of protons in the cell, which could cause cell death if acid extrusion is not induced to maintain homeostasis.

### 5.1. Glycolytic Metabolism

KLF4 and SP1 are involved in modulating the *SLC4A7/NBCn1* promoter, which encodes the Na^+^/HCO_3_^−^ cotransporter, NBCn1 [93]. The ErbB2 receptor tyrosine kinase stimulates this promoter, and KLF4 and SP1 were found to regulate downstream targets. Although KLF4 and SP1 belong to the SP/KLF superfamily, they have opposite effects on NBCn1 expression. KLF4 activates while SP1 inhibits NBCn1 levels (Figure 6(1A)). In another study, KLF4 behaved as an oncogene by stimulating glycolysis. KLF4 activates the phosphofructokinase platelet gene (*PFKP*) promoter to increase glucose uptake and lactate generation (Figure 6(1B)) [51]. The selection for the Warburg effect phenotype is seen in ductal carcinoma in situ (DCIS). Periluminal cells whose microenvironment is acidic, hypoxic, and nutrient-deficient have adapted to use glycolytic metabolism for energy. Subjecting low glycolytic breast cancer cells to harsh microenvironments (low glucose, low oxygen, and high acidity) caused the expression of the Warburg effect phenotype, which was indicated by a high ratio of extracellular acidification to oxygen consumption (ECAR/OCR). Spatial analysis of KLF4 in DCIS cells revealed that KLF4 upregulation occurred at the center of the duct where nutrients are sparse, acidosis is increased, and oxygen levels are decreased, suggesting that KLF4 may play a role in the switch to the Warburg phenotype [94].

Gorbatenko and colleagues showed that KLF4, which is frequently context-dependent, acted like an oncogene while SP1, which is usually elevated in cancer, was downregulated (Figure 6(1)) [93]. When studying the impact of ursolic acid on breast cancer development, SP1 had an inhibitory effect on glycolysis which was demonstrated by a decrease in lactate production and glycolysis-related protein expression (Figure 6(1C)). Treatment with ursolic acid increased SP1 expression and promoted SP1 binding to the Caveolin-1 (*CAV1*) promoter to increase *CAV1* transcription. Activation of CAV1 by SP1 downregulates glycolytic metabolism and damages mitochondrial function in breast cancer cells [95].

Furthermore, SP1 regulation of the hypoxia-inducible factor-1α (HIF-1α) transcription factor promotes glycolytic enzyme gene expression [96]. In breast cancer cells under hypoxic conditions, SP1 acts as a transcriptional activator. Hypoxia activates both SP1 and HIF-1α. The expression of glycolytic enzyme glyceraldehyde-3-phosphate dehydrogenase (GAPDH) was upregulated by the binding of HIF-1α to a hypoxia response element (HRE) when SP1 was also bound to the 3′ GC box flanking the HRE. The upregulation of GAPDH by HIF-1α depended on SP1 during hypoxia [97]. Insulin-stimulated SP1 and HIF-1α production and recruitment to the leptin promoter increased leptin expression. SP1 and HIF-1α activation by insulin allow for the regulation of glucose metabolism. Leptin overexpression in breast cancer cells promotes cell growth [98]. SP1 and HIF-1α regulate the thiamine transporter SLC19A3. SP1 binds the *SLC19A3* promoter during normoxia to allow for basal expression, and HIF-1α binds to the promoter during hypoxia to switch to an adaptive regulation. This allows the breast cancer cells to maintain thiamine homeostasis, which may support the Warburg phenotype as thiamine-dependent enzymes are crucial to cell metabolism [99].

### 5.2. Hexosamine Biosynthesis Pathway

In breast cancer, cancer stem-like cells (CSC) can alter energy metabolism to focus on the hexosamine biosynthesis pathway (HBP) to permit increased cell proliferation [100,101]. High KLF8 expression is associated with poor prognosis of breast cancer patients and was found to play a role in the HBP [102,103]. About 2–5% of glucose and glutamine transported into the cell enter the HBP to generate UDP-GlcNAc. O-GlcNAc transferase enzyme (OGT) adds O-GlcNAc moieties to proteins using the UDP-GlcNAc substrate. Increasing OGT and O-GlcNAc levels in breast CSCs cause increased glucose uptake, lipid metabolism, and glycolytic flux. KLF8 was increased in breast cancer cells overexpressing OGT (Figure 6(2)). Decreasing OGT levels decreased KLF8 and inhibited breast tumor growth [104].

### 5.3. Lipid Metabolism

SP1 plays a significant role in lipid metabolism [96,105]. In triple-negative breast cancer (TNBC) cell lines, SP1 is an activator of acyl-CoA synthetase 4 (ACSL4), which catalyzes the conversion of fatty acids to their active acyl-CoA form. SP1-mediated upregulation of *ACSL4* is associated with increased aggressiveness of TNBC tumors (Figure 6(3A)) [106].

In estrogen receptor-positive breast cancer, the most prevalent type of breast cancer, SP1 interacts with the circadian gene TIMELESS (TIM) to increase alkaline ceramidase 2 (ACER2) and enhance mitochondrial respiration. ACER2 is involved in the biosynthesis of S1P, an essential product of sphingolipid metabolism that was found to regulate mitochondrial oxidative phosphorylation. KEGG analysis showed that crucial proteins involved in the sphingolipid metabolism pathway were downregulated in *TIM* knockdown cells. TIM mediated sphingolipid metabolism and mitochondrial respiration through the SP1/ACER2/S1P axis [105] (Figure 6(3B)).

Glutamine deprivation can impede cancer development since glutamine can enter the tricarboxylic acid (TCA) cycle and participate in lipid metabolism. Some cancer cells rely on glutamine synthetase (GS) for de novo glutamine production. Glutamine can be broken down into acetyl CoA by oxidative glutamine metabolism. Acetyl CoA is converted to malonyl CoA by acetyl-CoA carboxylase 1 (ACC1), which is needed to form lipid droplets. Sterol regulatory element-binding protein 1 (SREBP1) is a lipogenic transcription factor that serves as an activator for GS. GS promotes the O-linked N-acetylglucosaminylation (OGlcNAcylation) of SP1 [107]. In the hexosamine biosynthesis pathway (HBP), glutamine is a precursor of UDP-GlcNAc for protein O-GlcNAcylation. Overexpression of GS increases O-GlcNAc-SP1 expression. O-GlcNAc-SP1 then enhances SREBP1 and ACC1 expression to increase lipogenesis and lipid droplet formation (Figure 6(3C)). GS thus uses the O-GlcNAc-SP1/SREBP1/ACC1 axis to regulate glutamine deprivation-induced LD formation and increase cell survival [107] (Figure 6(3C)). 

## 6. Brain and Nerve Tumors

Glioma, the most common type of brain tumor, has been divided into four groups based on malignancies [108]. Glioblastoma (GBM), also known as grade IV glioma, is one of the most aggressive forms of cancer [109].

### 6.1. Mitochondrial Fusion and Fission

In glioblastoma, abnormal cellular metabolism provides energy for increased cell proliferation. ATP synthase staining revealed that mitochondrial fusion is promoted in KLF4 expressing metabolically active cells, changing the mitochondrial morphology to tubular and interconnected [110]. Immunocytostaining was used to determine that KLF4 expression induces mitochondrial fusion and alters mitochondrial morphology by binding to methylated CpGs and interacting with guanine nucleotide exchange factor (GEF) family members. Fusion allows for increased ATP synthesis. During fission, on the other hand, the mitochondria become short and fragmented. Expression of KLF4 did not impact glucose uptake, glycolytic metabolism, glucose oxidation, or pentose pathway in normal conditions. When drugs that impaired mitochondrial function were administered, KLF4 enhanced the respiratory capacity of GBM cells. KLF4 expression in cells under stress caused an increase in oxygen consumption rate (OCR) and extracellular acidification rate (ECAR). Thus, in gliomas, KLF4 plays a role in recovering energy metabolism (Figure 7) [110,111].

### 6.2. Glycolytic Metabolism

Some GBM cell lines rely on glycolysis for energy over oxidative phosphorylation [112]. Overexpression of miR-181b has an inhibitory effect on glucose metabolism in GBM cells, while upregulation of SP1 has the opposite impact. The decrease in ECAR and glucose transporter protein GLUT1 showed that miR-181b suppressed glycolysis (Figure 7(1A)). By binding their promoters, SP1 activates GLUT1 and PKM2, which enhances glucose metabolism [113]. In an alternate study, the activation of KLF9 by G protein-coupled receptor 17 (GPR17) suppressed glioma development. GPR17 mediated KLF9 expression through Ring Finger Protein 2 (RNF2). Upon *GPR17* overexpression, RNF2 inhibition via cAMP/PKA/NF-κB signaling allowed for increased KLF9 levels. KLF9 decreased the expression of *SOD1*, a protein needed for ROS clearance (Figure 7(1B)). Upregulation of GPR17 also decreased cAMP levels, indicating that glycolysis and lactate production were also decreased [108,114].

In GBM cells, ZBTB2 blocks activation of the RelA/p65 gene by inhibiting SP1 binding to a GC box of the RelA/p65 proximal promoter. RelA/p65 binds to peroxisome proliferator-activated receptor-γ coactivator1α (PGC1α) to repress PDK4, a pyruvate dehydrogenase (PDH) inhibitor. PDH is a crucial regulator of glucose metabolism as it is responsible for converting pyruvate to acetyl CoA. ZBTB2 inhibition of RelA/p65 is accompanied by an increase in PDK4 expression and pyruvate and lactate levels displaying a shift toward glycolytic metabolism [115]. SP1 was found to regulate HIF-1α levels in glioma cells. Hypoxia-induced SP1 expression led to increased activation of disintegrin and metalloproteinase-17 (*ADAM17*) promoter, enhancing glioma’s invasiveness under hypoxic conditions. SP1 induction by hypoxia also caused the upregulation of HIF-1α, while inhibition of SP1 reduced HIF-1α levels [116]. Gliomas have elevated monoamine oxidase B (MAOB), SP1, and HIF-1α levels. HIF-1α is in charge of reprogramming the cell to depend on glycolysis for energy. In GBM cells under hypoxic conditions, SP3 binding is diminished, allowing for the activation of the genes it represses. MAOB produces hydrogen peroxide, which increases SP1 levels and decreases SP3 levels. The resulting ratio change of SP1 to SP3 causes the upregulation of HIF-1α [117].

### 6.3. Glycosaminoglycans Synthesis

KLF4 participates in the formation of glycosaminoglycans (GAGs) in GBM cells. KLF4 activates UDP-glucose 6-dehydrogenase (UGDH) expression by binding to methylated CpGs (mCpGs) in cis-regulatory elements (Figure 7(2,3)). UGDH is a rate-limiting enzyme in GAG monosaccharide synthesis, but it also regulates cell migration and proliferation. In the GAG synthesis pathway, glucose is converted to G6P to G1P to UDP-Glucose. UGDH then catalyzes the oxidation of UDP-Glucose to UDP-glucuronic acid (UDP-GlcA). Upregulation of GAG formation can support brain cancer progression. UGDH is overexpressed in GBM cells, and its expression correlates with that of KLF4 [118].

### 6.4. Lipid Metabolism

In GBM, SP1 was found to strengthen the resistance to the chemotherapy agent temozolomide (TMZ) [119]. It does this by increasing the expression of cytochrome p450 (CYP) 17A1, which is responsible for catalyzing the conversion of cholesterol to neurosteroids. Although the Warburg phenotype is shown in many cancers, some GBM cell lines rely on oxidative phosphorylation and fatty acid B oxidation (FAO) for energy. In GBM cells, SP1 increases prostaglandin E2 (PGE2), increasing mitochondrial activity to provide further resistance to TMZ. Prostaglandin-endoperoxide synthase 2 (PTGS), also known as cyclooxygenase (COX), is responsible for PGE2 synthesis and arachidonic acid (AA) metabolism (Figure 7(3A)). *SP1* knockdown caused a decrease in phospholipid metabolism-related proteins and AA-derived metabolites, proving SP1’s role in regulating the reprogramming of metabolism to gain TMZ resistance. SP1 modulates PGE2 expression after mitochondrial damage by TMZ to promote FAO and TCA cycle progression, increasing mitochondrial ATP production to sustain cell survival [120].

Neuroblastoma (NB) is a cancer that forms in nerve tissue [121]. Although some cancers can become dependent on lipid metabolism, NB cells experience insufficient lipid metabolism. Elongation of very long-chain fatty acids protein 4 (ELOVL4) catalyzes a rate-limiting step in the biosynthesis of very-long polyunsaturated fatty acids and modulates lipid droplet accumulation. SP1, along with the MYCN and histone deacetylases HDAC1 and HDAC2, binds the ELOVL4 promoter to suppress its expression and enhance cancer development (Figure 7(3D)). Increased ELOVL4 expression is associated with a better prognosis in patients as it can help to recover the LD deficiency in NB cells [122].

### 6.5. Oxidative Phosphorylation

The high demand for energy by neuronal cells is fueled by the oxidative pathway [123]. Nuclear respiratory factor 1 (NRF1) and SP factors were previously found to regulate the last enzyme in the respiratory electron transport chain, cytochrome c oxidase (COX), whose expression is crucial for energy production. NRF1 activates the *Gria2* (GluA2) gene of alpha-amino-3-hydroxy-5-methyl-4-isoxazolepropionic acid (AMPA) receptor. Gria2 responds to alterations of neuronal activity and COX expression. SP4 binding of the *Gria2* promoter upregulated the promoter’s activity (Figure 7(3C)). SP4 and NRF1 work together to couple energy metabolism and neuronal activity by regulating *Gria2* expression [124].

## 7. Metabolic Heterogeneity of Cancers

The current state of knowledge shows that even tumors that develop within the same organ or tissue can present high metabolic heterogeneity, creating variable metabolic phenotypes [125]. Dysregulated and reprogrammed metabolism promotes cancer progression and supports anabolic and pro-proliferative metabolism. Among factors creating metabolic heterogeneity, both intrinsic to cancer (e.g., genetic and epigenetic alterations) and micro environmentally derived (like a nutrient limitation, interactions with matrix, stroma, or immune cells) can be found [126]. Examples for both categories can be found in gastrointestinal cancers. Significantly, metabolic heterogeneity of cancers affects therapeutic strategy and clinical prognosis. Aberrant expression of genes and proteins in one tissue may alter cellular metabolism to favor oncogenesis and cancer progression. In contrast, the same gene/protein may have tumor-suppressive effects in other tissues. For example, as previously illustrated, silencing of *KLF4* suppresses *MGLL* expression resulting in enhanced proliferation in HCC [75] and colorectal cancers [127,128]. On the other hand, MGLL is upregulated in ovarian, breast, and prostate cancers and melanoma [129]. Similarly, increased KLF13 promotes ACOT7 expression leading to increased C18:1 free fatty acid synthesis in HCC [76]. In prostate cancer, however, silencing of KLF13 activates the AKT pathway and proliferation [130].

## 8. Conclusions

The current review highlights the role of SP and KLF transcription factors in cancer metabolism. Specifically, we described their role in gastrointestinal (esophageal, colon, liver, pancreas), breast, and brain cancers to emphasize the progress in the studies concerning metabolism regulation during carcinogenesis within the past decade. Presented examples of SP/KLF factors in modulating different metabolic pathways allow appreciating the complexity of the regulatory network. SP/KLF family regulates glucose and lipid metabolism, hexosamine biosynthesis pathway, oxidative phosphorylation, calcium levels, and mitochondria fission and fusion. This review only provides a snapshot into the role of one family of transcription factors in cancer metabolism. The relationship between SP/KLF factors and metabolism is complicated. It is essential to recognize that SP/KLF factors can exhibit opposite functions during homeostasis, and thus, their impact could be even more accentuated during carcinogenesis. Furthermore, insights into the role of SP/KLF factors could reveal the role of their upstream regulators and downstream targets in regulating metabolic pathways during cancer development and progression and improve our understanding of this complex process.

## Figures and Tables

**Figure 1 ijms-23-09956-f001:**
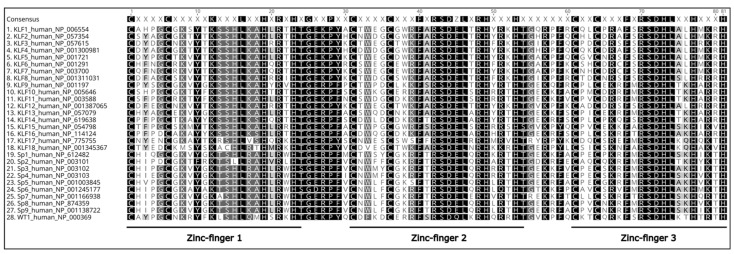
Sequence alignment of zinc-finger domains of SP and KLF proteins identified in humans (*Homo sapiens*). Amino acids in black are highly conserved, and similar residues are shown on a lighter background. The DNA binding domain of Wilms’ Tumor 1 (WT1) is included as an example of canonical zinc fingers. All sequences were obtained from the NCBI human genome database [20], assembled using MAAFT [36], and aligned in Geneious 10.2.6 software (https://www.geneious.com, accessed on 17 August 2022).

**Figure 3 ijms-23-09956-f003:**
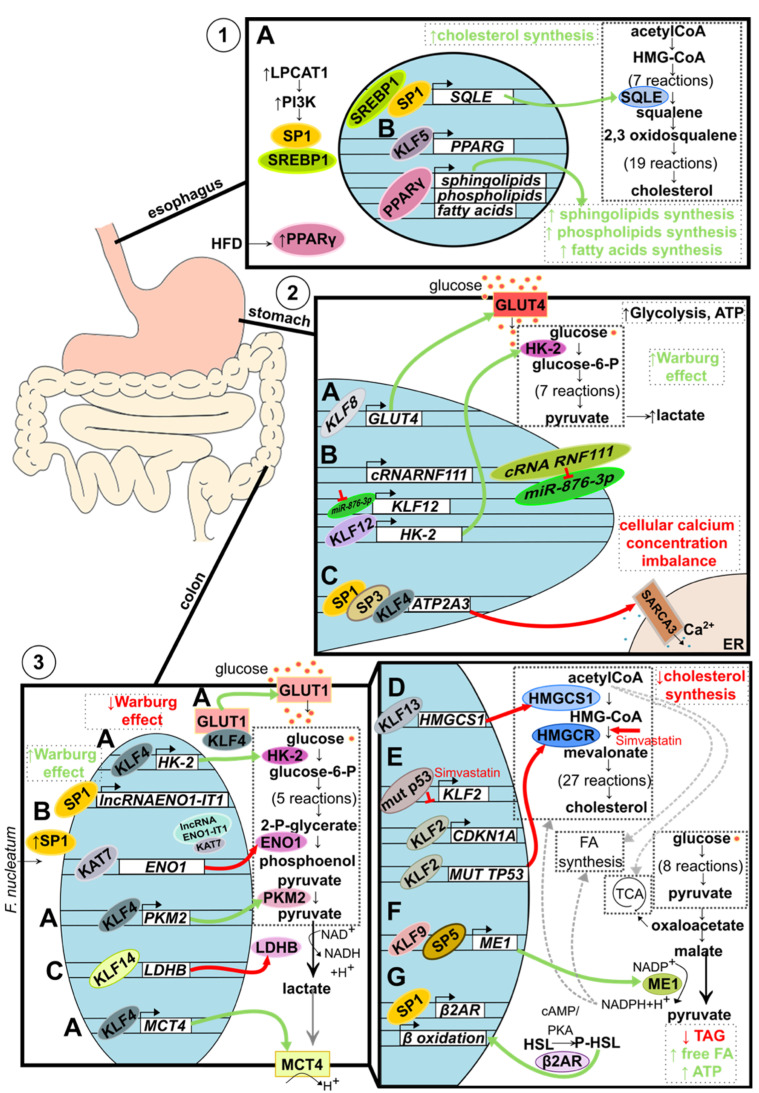
Metabolic alteration in GI track-associated cancer. (**1**) Alteration of lipid metabolism in esophageal cancer. (A) Increased level of LPCAT1 activates PI3K signaling pathways, which leads to SP1 and SREBP1 recruitment into the nucleus. Both transcription factors bind to the *SQLE* regulatory element and induce its expression resulting in increased de novo cholesterol synthesis. (B). Elevated during the tumorigenesis, KLF5 binds to the enhancer and promoter regions of *PPARG*, activating its expression. PPARγ then binds to the promoters of sphingolipids, phospholipids, and fatty acids synthesis-related genes resulting in increased de novo synthesis. Additionally, the level of PPARγ is stimulated environmentally by a high-fat diet (HFD). (**2**) Alteration of glucose metabolism in gastric cancer. (A) Upregulated KLF8 binds to the *GLUT4* promoter stimulating its expression and, as a consequence, increasing glucose uptake. (B) Interaction between circular RNA RNF111 and its target, miR-876-3p, leads to decreased ability of miR-876-3p to downregulate *KLF12* expression. Consequently, upregulated KLF12 stimulates HK-2 expression leading to increased lactate and ATP production. Additionally, KLF12 is believed to positively affect glucose uptake, leading to an increased Warburg effect. (C) SP1, SP3, and KLF4 collectively bind to the *ATP2A3* proximal promoter downregulating its expression. Lowered level of sarco/endoplasmic reticulum Ca^2+^ ATPase SERCA3 results in a loss of intracellular Ca^2+^ homeostasis and tumorigenesis suppression. (**3**) Alteration of glucose and lipids metabolism in colorectal cancer. (A) KLF4 decreases the Warburg effect by acting as a tumor suppressor, affecting glucose metabolism on multiple levels. KLF4 binds to the promoter region and upregulates the expression of key glycolytic enzymes: HK-2 and PKM2. Additionally, KLF4 upregulates the expression of lactate transporter *MCT4* and stimulates translocation of GLUT1 into the cell membrane. By doing so, KLF4 stimulates the overall glucose uptake and oxidative glucose metabolism and prevents lactic acid buildup. (B) Microbiota component, *F. nucleatum*, increases intracellular levels of SP1, leading to the induction of SP1-dependent lncRNA-ENO1-IT1 expression and histone acetyltransferase KAT7 recruitment. KAT7 changes the availability of the *ENO1* gene, regulates its expression, and downregulates glycolysis. (C) KLF14 binds to the *LDHB* promoter downregulating its expression. (A–C) Taken together, KLFs and SPs, in the case of colorectal cancer, act as tumor suppressors by turning glucose metabolism into less Warburg effect-like. (D) KLF13 binds to the *HMGCS1* promoter and downregulates its expression resulting in decreased de novo cholesterol synthesis. (E) Similarly, KLF2 reduces de novo cholesterol synthesis by mediating the simvastatin effect on HMGCR. A mutated variant of p53 protein present in 50% of colorectal cancer cases reduces *KLF2* expression, which leads to the downregulation of p21 protein levels. However, upon simvastatin treatment, the KLF2 level increases and upregulates *CDKN1A* expression, and downregulates the expression of the mutated variant of *TP53*, collectively resulting in decreased de novo cholesterol synthesis. (F) KLF9, together with SP5, increases *ME1* expression, gene encoding enzyme linking catabolic and anabolic metabolic pathways through NADPH^+^H^+^ and leads to an increased de novo synthesis of fatty acids and cholesterol. (G) SP1-dependent expression of *β2AR* results in an increased phosphorylation of HSL and consequently increased expression of β oxidation-related genes. As a result, the level of triglycerides is reduced, while the levels of free fatty acids and ATP increase.

**Figure 4 ijms-23-09956-f004:**
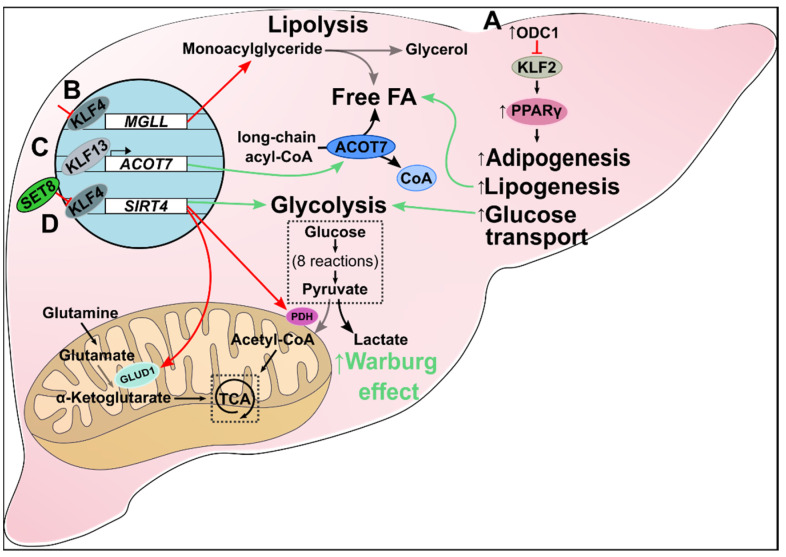
Metabolic alteration in liver cancer. (A) Increased ODC1 inhibits KLF2 expression, which upregulates PPARγ and downstream pathways. The result is increased de novo adipogenesis, lipogenesis, fatty acid accumulation, and glucose transport in HCC. (B) Inhibition of KLF4 suppresses transcription of *MGLL*, leading to dysregulated lipolysis and increased cellular monoacylglyceride levels. (C) KLF13 is a transcriptional promoter of *ACOT7*. In HCC, increased KLF13 upregulates *ACOT7* expression which drives the metabolism of long-chain acyl-CoAs to free monounsaturated fatty acids and CoAs. Additionally, C18:1 oleic acid (a monounsaturated fatty acid) production increases cell proliferation and migration. (D) KLF4 is a transcriptional promoter of *SIRT4*. SET8 inhibits KLF4, which, in turn, suppresses SIRT4 expression. Loss of SIRT4 reduces glutamate and pyruvate metabolism via decreased GLUD1 and PDH, respectively. Consequently, oncogenic cells obey the Warburg effect by shifting to aerobic glycolysis.

**Figure 5 ijms-23-09956-f005:**
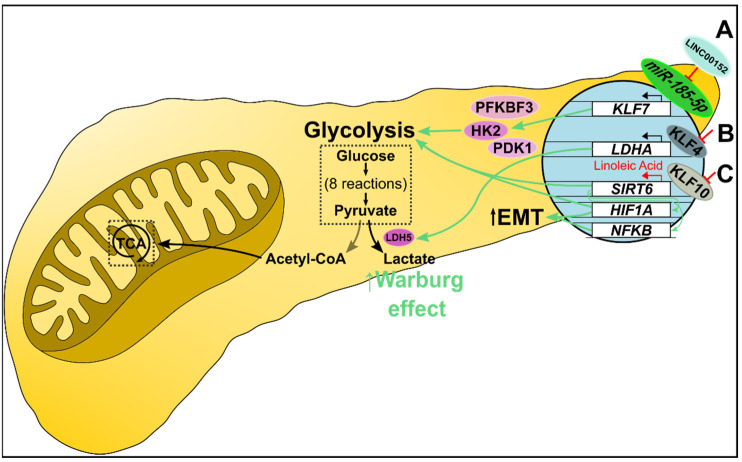
Metabolic alteration in pancreatic cancer. (A) miR-185-5p transcriptionally suppresses *KLF7* expression. In pancreatic cancer, overexpressed LINC00152 competitively binds miR-185-5p, which promotes *KLF7* transcription. KLF7 upregulates several glycolysis-related proteins, including HK2, PFKBF3, and PDK1, increasing glucose uptake, glycolysis, and lactate production. (B) KLF4 negatively regulates *LDHA* transcription. Decreased KLF4 promotes *LDHA* transcription and downstream expression of LDH M subunits. Consequently, LDH5 (a tetramer of M subunits) preferentially catalyzes pyruvate to lactate, essentially shunting cells to the Warburg effect. (C) KLF10 regulates glycolysis as a transcriptional promoter of *SIRT6*. Decreased KLF10 suppresses *SIRT6* transcription resulting in NFκB and HIF1α-mediated increased glycolysis and epithelial-mesenchymal transition. Furthermore, linoleic acid treatment increased SIRT6 levels and restored normal glucose metabolism in vitro, and increased survival in vivo.

**Figure 6 ijms-23-09956-f006:**
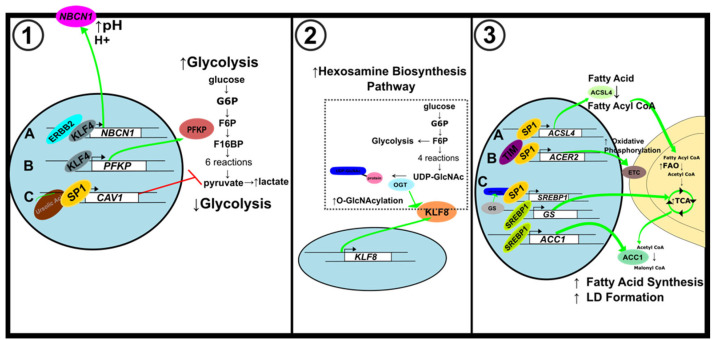
Metabolic alterations in breast cancer. (**1**) Alterations to glycolytic metabolism. (A) KLF4 activates *NBCN1* expression to maintain pH imbalance caused by increased glycolysis. (B) KLF4 increases glycolytic metabolism by activating PFKP expression. (C) Ursolic acid induces *CAV1* expression through SP1 to inhibit glycolysis. (**2**) Modifications to hexosamine biosynthesis pathway. KLF8 expression is upregulated by OGT, a key regulator of HBP. (**3**) Alterations to lipid metabolism. (A) SP1 activates *ACSL4* expression to increase FAO. (B) SP1 binds to the *ACER2* promoter to elevate mitochondrial oxidative phosphorylation. (C) The feedforward loop of O-GlcNAc-Sp1/SREBP1/ACC1 signaling enhances LD formation.

**Figure 7 ijms-23-09956-f007:**
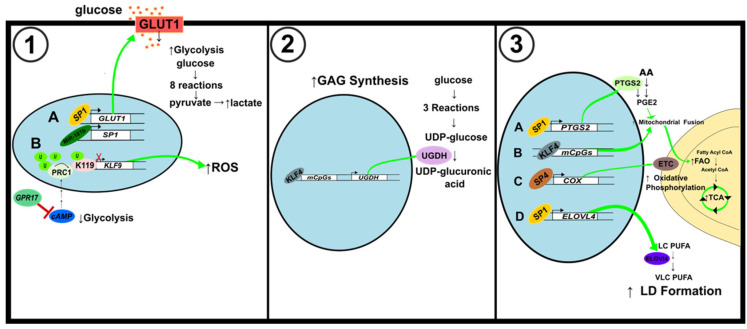
Schematic illustration of metabolic alterations in brain and nerve cancer. (**1**) Alterations to glycolytic pathway. (A) SP1 activation by miR-181b results in an increase in GLUT1 levels and ultimately glycolysis levels. (B) GPR17 inhibits cAMP to decrease PRC1-mediated histone H2A K119 monoubiquitination of the *KLF9* promoter. Activation of KLF9 increases ROS but reduces cell proliferation. (**2**) Alterations of glycosaminoglycans synthesis. KLF4 binding to mCpGs increases the expression of UGDH, a key regulator of GAG synthesis. (**3**) Modifications to mitochondrial fusion and fission, and lipid metabolism. (A) SP1 activates *PTGS2* to induce mitochondrial fusion and increase ATP production through FAO and the TCA cycle. (B) KLF4 binding to mCpGs also induces mitochondrial fusion. (C) SP4 activates the expression of *COX*, also known as Complex IV of the ETC, to increase mitochondrial oxidative phosphorylation. (D) SP1 binds to the *ELOVL4* promoter to augment LD formation.

## Data Availability

Not applicable.

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
