# Peer review of "SP and KLF Transcription Factors in Cancer Metabolism"

_ijms, 2022, doi:10.3390/ijms23179956_

Round 1

Reviewer 1 Report

Authors have comprehensively described the role of SP and KLF transcription in cancer metabolism. But the below mentioned points needs to be included before the manuscript is considered for publication in IJMS.

1. Author should mention the structural relationship of KLF. The relationship between all mammalian KLFs will also enrich the knowledge of readers about KLF. A pictorial representation will work for both.

2. Author should have also focused on Sp transcription factors in the regulation of HK (hexokinase), PK (pyruvate kinase), LDH (lactate dehydrogenase), and HIF (hypoxia-inducible factor-1α) in different cancer cell metabolism

Author Response

We would like to thank the Reviewer for his/her comments.

Comment 1. Author should mention the structural relationship of KLF. The relationship between all mammalian KLFs will also enrich the knowledge of readers about KLF. A pictorial representation will work for both.

Response 1. To address the Reviewer comment we added two new figures to the manuscript. Figure 1 portraits the similarity of the zinc-finger domains in SP and KLF proteins in humans while Figure 2 shows a phylogenetic tree that includes all SPs and KLFs proteins from several primates and rodent’s species. In addition, we cited several publications that include in-depth SP/KLF proteins structure analysis.

Comment 2. Author should have also focused on Sp transcription factors in the regulation of HK (hexokinase), PK (pyruvate kinase), LDH (lactate dehydrogenase), and HIF (hypoxia-inducible factor-1α) in different cancer cell metabolism.

Response 2. To address this comment, we provided information regarding SP1 in the context of hexokinase, pyruvate kinase, lactate dehydrogenase, and hypoxia-inducible factor-1α in the metabolism of liver, pancreas, breast, and brain tumors.

In addition, we modified the text to improve English language and style.

Reviewer 2 Report

The well-written and well-organized review about the impact of SP and KLF transcription factors in cancer metabolism. I have no major comments to improve this work. However, I think the only portion which is lacking in the current version of MS is "Metabolic Heterogeneity of Cancers", which the Authors did not touch on an important topic at all. I encourage the Authors to briefly explain the contribution of SP and KLF to the metabolic heterogeneity of cancers.

Author Response

We would like to thank you the Reviewer for favorable comments.

Comment 1. I think the only portion which is lacking in the current version of MS is "Metabolic Heterogeneity of Cancers", which the Authors did not touch on an important topic at all. I encourage the Authors to briefly explain the contribution of SP and KLF to the metabolic heterogeneity of cancers.

Response 1. To address this comment, we provided a short summary (new section 7) regarding metabolic heterogeneity of cancers.

In addition, we modified the text to improve English language and style.

Round 2

Reviewer 1 Report

Authors have addressed the comment and improved the quality of this manuscript as well. This manuscript can be acceptable in present form.